# Learning from Disjoint Views: A Contrastive Prototype Matching Network for Fully Incomplete Multi-View Clustering

**Yiming Wang**[1,2]     **Qun Li**[1,2]     **Dongxia Chang**[3,4]     **Jie Wen**[5]
**Hua Dai**[1,2]     **Fu Xiao**[1,2,*]     **Yao Zhao**[3,4]

[1]School of Computer Science, Nanjing University of Posts and Telecommunications
[2]State Key Laboratory of Tibetan Intelligence, Nanjing University of Posts and
Telecommunications     [3]Institute of Information Science, Beijing Jiaotong University
[4]Beijing Key Laboratory of Advanced Information Science and Network Technology
[5]School of Computer Science and Technology, Harbin Institute of Technology
ymwang@njupt.edu.cn, liqun@njupt.edu.cn, dxchang@bjtu.edu.cn,
jiewen_pr@126.com, daihua@njupt.edu.cn, xiaof@njupt.edu.cn,
yzhao@bjtu.edu.cn

## Abstract

Multi-view clustering aims to enhance clustering performance by leveraging information from diverse sources. However, its practical application is often hindered by a barrier: the lack of correspondences across views. This paper focuses on the understudied problem of fully incomplete multi-view clustering (FIMC), a scenario where existing methods fail due to their reliance on partial alignment. To address this problem, we introduce the Contrastive Prototype Matching Network (CPMN), a novel framework that establishes a new paradigm for cross-view alignment based on matching high-level categorical structures. Instead of aligning individual instances, CPMN performs a more robust cluster prototype alignment. CPMN first employs a correspondence-free graph contrastive learning approach, leveraging mutual $k$-nearest neighbors (MNN) to uncover intrinsic data structures and establish initial prototypes from entirely unpaired views. Building on the prototypes, we introduce a cross-view prototype graph matching stage to resolve category misalignment and forge a unified clustering structure. Finally, guided by this alignment, we devise a prototype-aware contrastive learning mechanism to promote semantic consistency, replacing the reliance on the initial MNN-based structural similarity. Extensive experiments on benchmark datasets demonstrate that our method significantly outperforms various baselines and ablation variants, validating its effectiveness.

## 1 Introduction

Over the recent years, the rapid increase in multi-view data has made effective analysis an urgent necessity[17, 15]. Multi-view clustering, which seeks to uncover underlying data relationships by partitioning instances into distinct categories without supervision, has emerged as a prominent solution. While these algorithms have advanced significantly and are widely applied in applications like recommendation systems and multimedia analysis [26, 38, 37, 29], most existing algorithms heavily rely on the assumption of complete multi-view information [3, 33]. However, in practice, it is

---
*

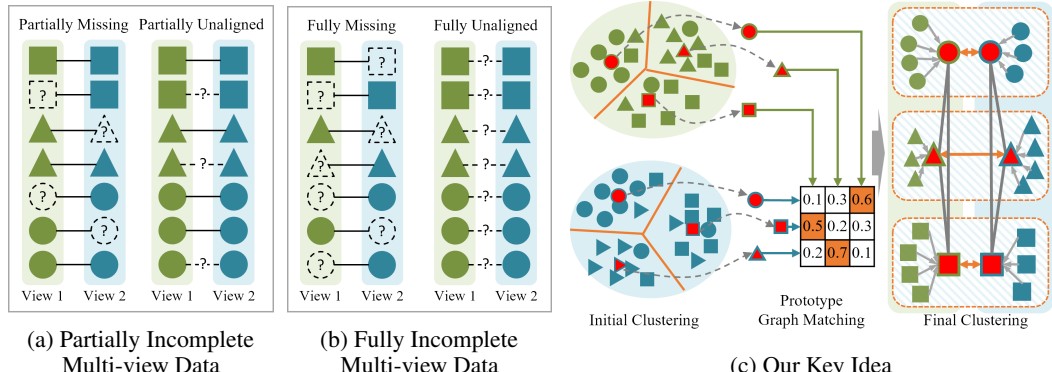

Figure 1: Various incomplete multi-view data and our key idea. Without loss of generality, data from two views are presented as an example. In the figure, different colors represent different views, and different shapes indicate distinct categories. The gray lines denote the availability of cross-view correspondences. The blank shapes and the dashed lines with question marks represent the missing instances and the missing correspondences between views, respectively. To achieve unified clustering for multi-view data without correspondence, our model first learns the initial cluster prototypes of each view and perform graph matching on the prototypes. Subsequently, the instances are assigned to the aligned prototypes to obtain the final clustering results.

common for only partial data to be collected or transmitted due to sensor failure, varying acquisition costs, or privacy concerns, making the assumption of data completeness difficult to satisfy.

To address the issue of incomplete data, many efforts have been made to explore ways of understanding multi-view data from partially incomplete information (Figure 1a), which is referred to as partially incomplete multi-view clustering (PIMC). The core mechanism of PIMC lies in leveraging available partial cross-view correspondences to establish informative priors for learning. Two principal approaches have arisen based on the differing conditions of data incompleteness. For multi-view data with partially missing views, existing methods attempt to exploit known cross-view correspondences to learn shared structures and recover missing information in the latent space [15, 1]. Alternatively, for partially aligned multi-view data, existing approaches aim to learn optimal transformations or permutation matrices that align the different view-specific spaces, establishing links between matching views [35, 28].

While these methods successfully mitigate the impact of partially incomplete information, their effectiveness fundamentally relies on the availability of at least partial cross-view information or alignment, which is often difficult to guarantee [41]. In many challenging real-world scenarios, data from different views is collected, processed, and stored entirely separately. Due to data heterogeneity, privacy regulations, or the nature of data acquisition, establishing correspondences even for a small subset of instances can be infeasible. This leads to the critical problem of fully incomplete multi-view data, where instances are characterized by data from only one view, and no cross-view correspondences are known a priori (Figure 1b). An example is in clinical diagnostics, where patients may undergo only one type of pathological examination, resulting in single-view information with no link between patients across different test types. Tackling this scenario requires learning shared cluster structures without any correspondences between views.

The Fully Incomplete Multi-view Clustering (FIMC) presents significant algorithmic challenges compared to PIMC. Unlike PIMC, where known correspondences provide direct guidance for imputation or alignment, the fully incomplete scenario lacks cross-view anchors that could guide representation alignment. Consequently, imputation approaches based on cross-view correspondences are inapplicable, and alignment methods relying on paired instances cannot be initialized or directly learned. Recently, a limited number of pioneering studies have started to tackle the FIMC challenge. Notably, SMILE [41] focuses on learning semantic invariant representations for FIMC. This approach effectively mitigates the absence of correspondences by ensuring that instances belonging to the same underlying cluster have similar representations regardless of their view. However, this approach assumes that semantic distributions are strictly invariant across views, which may fail under noisy views or class imbalance, leading to biased alignment. This motivates the exploration of methods

that explicitly handle the alignment of cluster prototypes and utilize category-level information for representation refinement once initial cluster structures emerge.

This paper proposes the novel Contrastive Prototype Matching Network (CPMN) to introduce a new paradigm of structure-level alignment to address FIMC. Instead of aligning instances or enforcing rigid distributional invariance, CPMN operates at a higher level of abstraction by aligning robust categorical prototypes ( (Figure 1c). Our approach first employs a correspondence-free cross-view graph contrastive learning strategy, utilizing mutual $k$-nearest neighbors (MNN) to learn initial representations and generate view-specific prototypes via K-means. Crucially, we then explicitly bridge the view-specific cluster structures by introducing a similarity-based prototype graph matching to find the optimal mapping between these initial prototypes. Finally, CPMN refines the representations and prototypes through a prototype contrastive learning mechanism, which leverages the established prototype correspondences and category information to contrast instances against their matched prototypes. The key contributions of our work include:

- We propose a novel framework, Contrastive Prototype Matching Network (CPMN), designed to address the challenging fully incomplete multi-view clustering problem where no cross-view correspondences exist.
- We introduce a two-stage contrastive learning approach: an initial correspondence-free graph contrastive learning phase based on cross-view MNN for structural alignment, followed by a prototype contrastive learning phase that utilizes category-level information derived from matched prototypes for semantic refinement.
- We incorporate an explicit prototype graph matching strategy to construct a bipartite graph between view-specific prototypes and find the optimal mapping between prototypes based on their feature similarity.

## 2   Related Works

Before introducing the proposed CPMN, we briefly review the related partially missing multi-view clustering and partially aligned multi-view clustering.

Partially Missing Multi-view Clustering (PMMC) aims to learn unified cluster assignments from multi-source data where some views are partially missing for certain instances. Among the methods that broadly employ either conventional shallow models [27, 18] or deep neural networks (DNN) [30, 1], our attention here is restricted to the DNN-based approaches. Owing to the capability of the autoencoder (AE) to handle missing data through feature reconstruction, prevailing methods generally adopt AE as their backbone. Some approaches combine AE with graph information to mitigate the shortcomings of structural information mining in AE. For example, SDIMC-net [31] combines graph embedding strategies to capture the high-level features and local structure of each view. Likewise, CASEN [34] performs adaptive graph convolution on the reconstructed multi-view data to effectively extract the data structure. Some approaches introduce contrastive learning [8, 2] to learn a shared representation that is robust to view missingness. DCP [15] performs data recovery and consistency learning simultaneously through dual contrastive prediction. Introducing deep subspace learning, PVC-SSN [4] employs contrastive learning to obtain more discriminative and consistent subspace representations. DCMVSC [40] combines contrastive learning and Cauchy-Schwarz divergence to propose a new multi-view deep subspace clustering approach. While existing approaches demonstrate improved robustness to incomplete data, their dependence on at least partial cross-view correspondences persists as a limitation, particularly in scenarios without alignment.

Partially Aligned Multi-view Clustering (PAMC) addresses the critical challenges of view misalignment and unknown correspondences caused by asynchronous data acquisition in multi-view clustering. Existing methods can be broadly classified into two categories. The first category focuses on explicit alignment learning, while the second category focuses on robust representation learning without explicit alignment. As a pioneering study, PVC [10] addresses the partial alignment problem by proposing a differentiable Hungarian network to find correspondences between two views. Distinctly, EGPVC [42] employs Dykstra's cyclic constraint projection algorithm for the same purpose. However, explicit instance-level alignment in such methods is sometimes difficult to achieve and is overly restrictive for clustering tasks. To learn robust representation without explicit alignment, MvCLN [36] devises a noise-robust contrastive loss to reduce the impact of false negative pairs on aligning data and representation learning. SiMVC [25] completely avoids explicit representation alignment

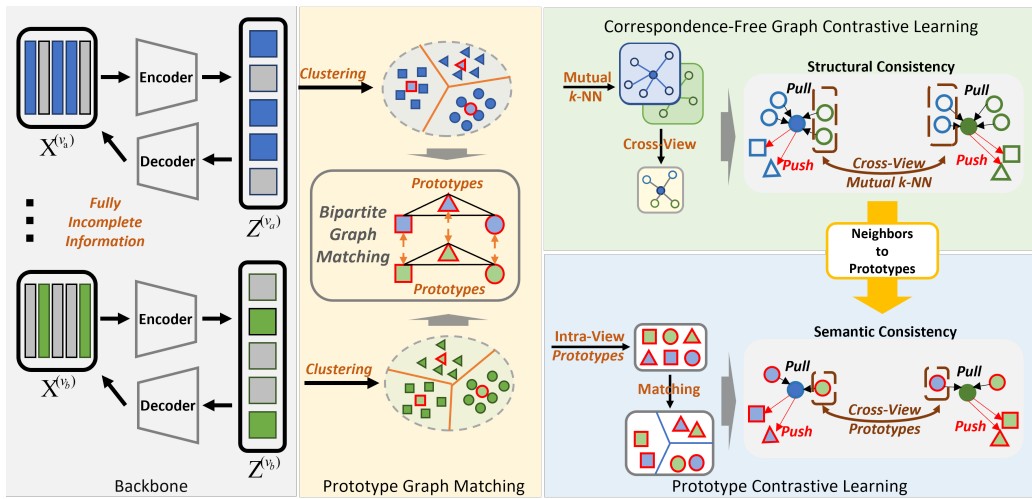

Figure 2: The framework of the proposed CPMN.

and instead uses learned linear combinations for view prioritization. The above methods achieve expected performance, but their alignment processes, which are often applicable only to two views or necessitate partial alignment information, consequently limit their broader applicability.

## 3 Method

In this section, we elaborate on the proposed Contrastive Prototype Matching network (CPMN) for learning a unified clustering partition from fully incomplete multi-view data. As illustrated in Fig. 2, CPMN consists of three novel modules, namely correspondence-free graph contrastive learning, prototype graph matching, and prototype contrastive learning, which will be detailed in Sec. 3.1-3.3

In fully incomplete multi-view clustering, paired information across views is absent. That is, all instances have features from only one view, or the correspondences between views are completely unknown. Formally, given a fully incomplete multi-view dataset $\mathcal{X} = \{X^{(v)}\}_{v=1}^V$ with $N$ instances in $V$ views. $X^{(v)} = \{x_1^{(v)}, ..., x_N^{(v)}\} \in \mathbb{R}^{N \times d_v}$ denotes the feature matrix in the $v$-th view, where $d_v$ is the feature dimension. For instances with only one view, we introduce an indicator matrix $W \in \{0, 1\}^{N \times V}$ to indicate the completeness of views, where $W_{i,v} = 1$ if instance $x_i$ exists in the $v$-th view, and $W_{i,v} = 0$ otherwise. Thus, for any instance $x_i$ of a fully incomplete dataset, $\sum_{v=1}^V W_{i,v} = 1$. For datasets without known correspondences, an indicator vector $U \in \{0, 1\}^N$ is used to record the aligned instances, where $U_i = 1$ means the $i$-th instance is aligned in all views, otherwise $U_i = 0$. Thus, for a fully incomplete dataset, $U_i = 0$ for all $i \in [1, N]$. Our goal is to partition these $N$ instances into $C$ clusters.

### 3.1 Correspondence-Free Graph Contrastive Learning

To effectively address the challenges of FIMC, the initial crucial step in our CPMN is to learn latent representations for each view independently, while simultaneously encouraging cross-view consistency without relying on paired data. This stage, termed correspondence-free graph contrastive learning, is designed to uncover underlying structural similarities across views. It achieves this by first reconstructing individual view data to preserve view-specific information, and then employing a novel graph-based contrastive learning strategy to promote coherence between the learned view-specific representations, thereby laying a foundation for subsequent prototype generation and matching.

A key component for learning these initial view-specific representations is the foundational backbone network: view-specific autoencoders. For each view $v$, the architecture comprises an encoder $E_v : \mathbb{R}^{d_v} \to \mathbb{R}^{d_z}$ and a corresponding decoder $D_v : \mathbb{R}^{d_z} \to \mathbb{R}^{d_v}$. The encoder maps the input data $x_i^{(v)}$ to a low-dimensional latent representation $z_i^{(v)} = E_v(x_i^{(v)})$, and the decoder attempts to reconstruct the original input $\hat{x}_i^{(v)} = D_v(z_i^{(v)})$. The model is trained by minimizing the weighted

reconstruction error for observed instances in each view, formulated as:

$$L_{REC} = \sum_v \sum_i W_{i,v} ||\hat{x}_i^{(v)} - x_i^{(v)}||_2^2 \tag{1}$$

While view-specific autoencoders can capture latent features within individual views, they do not inherently model cross-view relationships, resulting in divergent latent spaces that impede view alignment. Prevailing cross-view contrastive learning in multi-view clustering typically addresses this using cross-view instance pairs [16, 6, 22]. However, the absence of view correspondences in FIMC renders such pairwise instance selection infeasible. To overcome this limitation, we propose a correspondence-free graph contrastive learning strategy based on mutual $k$-nearest neighbors (MNN), which establishes pseudo-alignment through feature affinity without relying on prior correspondences. To learn discriminative representations while simultaneously mitigating cross-view discrepancies, CPMN involves the separate construction of intra-view and cross-view MNN graphs, which subsequently guide the establishment of positive pairs.

For intra-view consistency, positive pairs are identified based on the MNN criterion applied to the latent representations. Specifically, a pair of representations $(z_i^{(v)}, z_j^{(v)})$ from the same view $v$ is designated as positive if $z_j^{(v)}$ is found among the $k$-nearest neighbors of $z_i^{(v)}$, and conversely, $z_i^{(v)}$ is also present among the $k$-nearest neighbors of $z_j^{(v)}$. All other pairs within this view are treated as negative pairs. To quantify the affinity between these pairs, cosine similarity is utilized:

$$S_{ij}^{(v)} = \text{sim}(z_i^{(v)}, z_j^{(v)}) = \frac{(z_i^{(v)})^T z_j^{(v)}}{||z_i^{(v)}|| ||z_j^{(v)}||} \tag{2}$$

The objective is to pull together the representations of connected samples in the graph while pushing apart unconnected ones. Thus, the loss function can be formulated as:

$$L_{\text{SGL}}^{(v)} = -\sum_i \frac{1}{|\mathcal{P}_i^{(v)}|} \sum_{j \in \mathcal{P}_i^{(v)}} \log \frac{\exp(S_{ij}^{(v)}/\tau)}{\sum_{k \neq i} \exp(S_{ik}^{(v)}/\tau)} \tag{3}$$

where $\mathcal{P}_i^{(v)}$ is the set of indices of instances forming positive pairs with $z_i^{(v)}$ within view $v$, and $\tau$ is the temperature parameter.

Complementing the intra-view learning, cross-view positive pairs are established between different views, enabling alignment without correspondences. Considering a two-view example, positive pairs $(z_i^{(v_a)}, z_j^{(v_b)})$ are similarly established if they are MNNs of each other across views $v_a$ and $v_b$. All other pairings between these views are treated as negative pairs. Cosine similarity is again employed to measure the similarity, denoted as $S_{ij}^{(v_a,v_b)} = \text{sim}(z_i^{(v_a)}, z_j^{(v_b)})$. Cross-view graph contrastive learning aims to uncover and align latent structures, thereby achieving partial representation consistency despite the absence of correspondence. The loss function takes a similar form to the single-view contrastive loss:

$$L_{\text{CGL}}^{(v_a,v_b)} = -\sum_i \frac{1}{|\mathcal{P}_i^{(v_a,v_b)}|} \sum_{j \in \mathcal{P}_i^{(v_a,v_b)}} \log \frac{\exp(S_{ij}^{(v_a,v_b)}/\tau)}{\sum_k \exp(S_{ik}^{(v_a,v_b)}/\tau)} \tag{4}$$

where $\mathcal{P}_i^{(v_a,v_b)}$ is the set of indices of instances from view $v_b$ forming positive pairs with $z_i^{(v_a)}$.

The overall objective for this initial stage of correspondence-free graph contrastive learning, denoted as $\mathcal{L}_{\text{CFGC}}$, integrates the aforementioned losses:

$$\mathcal{L}_{\text{CFGC}} = L_{\text{REC}} + \lambda L_{\text{GL}}, \quad L_{\text{GL}} = \sum_v L_{\text{SGL}}^{(v)} + \sum_{v_a} \sum_{v_b \neq v_a} L_{\text{CGL}}^{(v_a,v_b)} \tag{5}$$

where $\lambda$ is a trade-off parameter between the two losses.

Following the representation learning guided by $\mathcal{L}_{\text{CFGC}}$, initial view-specific cluster structures are derived. Specifically, K-Means is applied to the latent representations $Z^{(v)}$ to obtain preliminary clustering assignments and a set of prototypes $\mathcal{C}^{(v)} \in \mathbb{R}^{C \times d_z}$. These prototypes are crucial for the subsequent graph matching.

## 3.2 Prototype Graph Matching

Following the correspondence-free contrastive learning, we successfully obtain view-specific representations and initial cluster prototypes for each view. However, these prototypes are independently derived and thus lack explicit cross-view correspondences. To bridge this gap, we devise a prototype graph matching strategy to establish correspondence to enable cross-view learning and fusion.

To establish an alignment between the independently derived sets of prototypes from different views (e.g., view $v_a$ and view $v_b$), we formulate this challenge as a principled optimization problem, specifically the classic Linear Assignment Problem (LAP), which can be modeled via bipartite graph matching. In our model, one set of nodes in the bipartite graph consists of the $C$ prototypes from view $v_a$, denoted $\mathcal{C}^{(v_a)} = \{c_1^{(v_a)}, \ldots, c_C^{(v_a)}\}$, and the other set comprises the $C$ prototypes from view $v_b$, $\mathcal{C}^{(v_b)} = \{c_1^{(v_b)}, \ldots, c_C^{(v_b)}\}$. An edge between a prototype $c_i^{(v_a)}$ and a prototype $c_j^{(v_b)}$ represents a potential correspondence, and the core task is to find the most plausible one-to-one mapping between these two sets.

To quantify the suitability of each potential match, we construct a cost matrix $M^{(v_a, v_b)} \in \mathbb{R}^{C \times C}$, where $M_{ij}^{(v_a, v_b)}$ represents the cost of matching the $i$-th prototype from view $v_a$, with the $j$-th prototype from view $v_b$. This cost is based on the dissimilarity between prototypes, which is formulated as:

$$M_{ij}^{(v_a, v_b)} = \frac{1}{\exp(\mathrm{sim}(c_i^{(v_a)}, c_j^{(v_b)}))}, \quad \mathrm{sim}(c_i^{(v_a)}, c_j^{(v_b)}) = \frac{(c_i^{v_a})^\top c_j^{v_b}}{\|c_i^{v_a}\| \|c_j^{v_b}\|} \tag{6}$$

where $c_i^{v_a}$ denotes the $i$-th prototype from view $v_a$.

The goal is to find an optimal permutation of indices, $\pi : \mathcal{C}^{(v_a)} \to \mathcal{C}^{(v_b)}$, that establishes a one-to-one mapping. This mapping assigns each prototype $c_i^{(v_a)}$ in view $v_a$ to a unique prototype $c_{\pi(i)}^{(v_b)}$ in view $v_b$, such that the cumulative cost of all matches is minimized. This objective can be expressed as:

$$\min_{\pi} \sum_{i=1}^{C} M_{i, \pi(i)} \tag{7}$$

This formulation guarantees that the problem is solvable in polynomial time. We find the globally optimal solution using the efficient Hungarian algorithm [13]. The output of the algorithm is the optimal matching, providing a mapping from the prototype indices of view $v_a$ to those of view $v_b$.

The established correspondence between prototypes creates a vital bridge for transferring clustering information across views. This enables us to incorporate cross-view category information into contrastive learning, using the category information in one view to guide the clustering in another.

## 3.3 Prototype Contrastive Learning

Leveraging the globally optimal prototype alignment, this stage transitions the learning objective from structure discovery to semantic refinement. It utilizes the matched prototypes as robust categorical anchors to refine both representations and prototypes. By contrasting instances with these anchors, this module elevates the alignment objective from the initial MNN-based structural similarity to a higher-order, globally consistent semantic alignment in the latent space. Additionally, for dynamic updates to cluster assignments and prototypes, a learnable clustering layer is used to replace the previous K-means methodology.

This prototype contrastive learning leverages the prototype mapping $\pi : \mathcal{C}^{(v_a)} \to \mathcal{C}^{(v_b)}$ established previously to guide representation refinement. For a given representation $z_i^{(v_a)}$, let $j$ be the index of the prototype $c_j^{(v_a)}$ to which it is assigned within view $v_a$. The core idea is to encourage $z_i^{(v_a)}$ to be simultaneously similar to its assigned prototype $c_j^{(v_a)}$ and to the corresponding prototype in view $v_b$,

$c_{\pi(j)}^{(v_b)}$. This objective is realized through two complementary loss components, defined as:

$$L_{\text{SPL}}^{(v)} = -\sum_i \log \frac{\exp(Sc_{ij}^{(v)}/\tau)}{\sum_{c=1}^{C} \exp(Sc_{ic}^{(v)}/\tau)}, \tag{8}$$

$$L_{\text{CPL}}^{(v_a,v_b)} = -\sum_i \log \frac{\exp(Sc_{ij}^{(v_a,v_b)}/\tau)}{\sum_{c=1}^{C} \exp(Sc_{ic}^{(v_a,v_b)}/\tau)} \tag{9}$$

Here, for instance $z_i^{(v)}$, the index $j$ in the numerators $Sc_{ij}^{(v)}$ and $Sc_{ij}^{(v_a,v_b)}$ refers to its assigned prototype index for the positive term. Thus, $L_{\text{SPL}}^{(v)}$ promotes the proximity of each representation $z_i^{(v)}$ to its assigned prototype $c_j^{(v)}$ within view $v$. Concurrently, $L_{\text{CPL}}^{(v_a,v_b)}$ fosters alignment by encouraging $z_i^{(v_a)}$ from view $v_a$ to be closer to the cross-view corresponding prototype $c_{\pi(j)}^{(v_b)}$ in view $v_b$. The terms $Sc_{ij}^{(v)} = \text{sim}(z_i^{(v)}, c_j^{(v)})$ and $Sc_{ij}^{(v_a,v_b)} = \text{sim}(z_i^{(v_a)}, c_{\pi(j)}^{(v_b)})$ represent the cosine similarity between representations and prototypes, within or across views respectively.

To further enhance cluster coherence and embedding discriminability, a self-supervision strategy inspired by [7] is integrated within the learnable clustering layers. For each instance $z_i^{(v)}$, $q_{ik}^{(v)}$ denotes its soft assignment to prototypes $c_k^{(v)}$, which is computed using the Student's t-distribution [20] as:

$$q_{ik}^{(v)} = \frac{(1 + ||z_i^{(v)} - c_k^{(v)}||_2^2)^{-1}}{\sum_{j=1}^{C}(1 + ||z_i^{(v)} - c_j^{(v)}||_2^2)^{-1}} \tag{10}$$

Subsequently, an auxiliary target distribution $p_{ik}^{(v)}$ is formulated by sharpening $q_{ik}^{(v)}$ as:

$$p_{ik}^{(v)} = \frac{(q_{ik}^{(v)})^2/\sum_j q_{jk}^{(v)}}{\sum_{k'}\left((q_{ik'}^{(v)})^2/\sum_j q_{jk'}^{(v)}\right)} \tag{11}$$

The clustering refinement is then driven by minimizing the Kullback-Leibler (KL) divergence between the current soft assignment distribution $Q^{(v)}$ and the target distribution $P^{(v)}$ for each view:

$$L_{\text{CLU}} = \sum_v \sum_i D_{\text{KL}}(P_i^{(v)}||Q_i^{(v)}) \tag{12}$$

where $P_i^{(v)}$ and $Q_i^{(v)}$ are the target distribution and current soft assignment vectors for instance $z_i^{(v)}$, respectively, and $D_{\text{KL}}$ denotes the KL divergence.

The overall optimization objective for the prototype contrastive learning stage is a weighted sum of the clustering loss and the prototype contrastive losses:

$$\mathcal{L}_{\text{PCL}} = L_{\text{CLU}} + \gamma L_{\text{PL}}, \quad L_{\text{PL}} = \sum_v L_{\text{SPL}}^{(v)} + \sum_{v_a} \sum_{v_b \neq v_a} L_{\text{CPL}}^{(v_a,v_b)} \tag{13}$$

where $\gamma$ is a trade-off parameter between the two losses. Joint optimization of these terms produces discriminative representations and refined prototypes, culminating in a robust clustering solution for fully incomplete multi-view data.

## 4 Experiments

In this section, we conduct experiments on five popular multi-view datasets to evaluate the effectiveness of the proposed CPMN.

### 4.1 Experimental Settings

**Datasets** Five widely-used datasets are used to evaluate the proposed CPMN. (1) **Caltech-101** is an image dataset containing 8677 instances from 101 categories. Following [41], deep features extracted by DECAF [12] and VGG19 [21] are used as two views. (2) **Cub** includes several categories of

Table 1: Fully incomplete multi-view clustering performance comparison on five widely-used multi-view datasets. "-" denotes that the baselines are not scalable to large datasets, The best and second best results are shown in **bold** and underlined, respectively.

| Data Type | Method | Caltech-101 | | | Cub | | | NoisyMNIST | | | MNIST-USPS | | | YouTubeFaces | | |
|---|---|---|---|---|---|---|---|---|---|---|---|---|---|---|---|---|
| | | ACC | NMI | ARI | ACC | NMI | ARI | ACC | NMI | ARI | ACC | NMI | ARI | ACC | NMI | ARI |
| 100% Unaligned (ζ = 100%) | MVC-UM | 43.3 | 67.3 | 31.9 | 44.3 | 40.7 | 23.0 | 19.3 | 9.9 | 4.7 | 53.5 | 48.4 | 35.0 | – | – | – |
| | GWMAC | 4.9 | 16.0 | 0.3 | 28.3 | 21.0 | 9.1 | 11.4 | 0.3 | 0.1 | 15.6 | 3.7 | 1.5 | 3.2 | 2.3 | 0.2 |
| | UPMGC-SM | 51.0 | 73.4 | 33.7 | **74.0** | 69.8 | **59.7** | 52.6 | 48.8 | 36.8 | 59.8 | 58.5 | 45.5 | – | – | – |
| | SMILE | 47.6 | 74.0 | 33.0 | 63.4 | 61.9 | 48.2 | 82.7 | 79.5 | 74.2 | 85.2 | 80.8 | 76.1 | 52.5 | 73.6 | 42.6 |
| | CPMN | 57.0 | 77.0 | 68.4 | 72.5 | 72.1 | 55.2 | 84.8 | 80.2 | 79.8 | 88.0 | 82.1 | 80.8 | 70.0 | 82.5 | 57.5 |
| 100% Missing (ψ = 100%) | DM2C | 28.2 | 59.3 | 18.3 | 35.6 | 36.4 | 6.4 | 23.2 | 15.4 | 8.0 | 35.1 | 34.2 | 18.3 | 16.2 | 32.1 | 5.8 |
| | SMILE | 30.5 | 60.1 | 20.4 | 40.2 | 37.5 | 20.8 | 69.0 | 63.8 | 54.1 | 74.3 | 69.6 | 61.8 | 26.5 | 49.9 | **18.5** |
| | CPMN | 38.7 | 68.2 | 40.7 | 47.0 | 49.8 | 29.6 | 71.1 | 68.2 | 58.8 | 75.5 | 69.2 | 66.0 | 27.4 | 51.3 | 16.2 |

Table 2: Partially incomplete multi-view clustering performance comparison on five widely-used multi-view datasets. "-" denotes that the baselines are not scalable to large datasets, The best and second best results are shown in **bold** and underlined, respectively.

| Data Type | Method | Caltech-101 | | | Cub | | | NoisyMNIST | | | MNIST-USPS | | | YouTubeFaces | | |
|---|---|---|---|---|---|---|---|---|---|---|---|---|---|---|---|---|
| | | ACC | NMI | ARI | ACC | NMI | ARI | ACC | NMI | ARI | ACC | NMI | ARI | ACC | NMI | ARI |
| 50% Unaligned (ζ = 50%) | PVC | 18.6 | 48.9 | 14.6 | 50.2 | 56.3 | 38.6 | 81.8 | 82.3 | 82.0 | 86.5 | 78.1 | 74.6 | – | – | – |
| | MvCLN | 35.6 | 61.0 | 40.9 | 58.2 | 55.2 | 40.8 | 91.1 | 84.2 | 83.6 | 90.0 | 81.4 | 80.4 | 54.0 | 69.2 | 44.2 |
| | SURE | 46.2 | 70.7 | 33.0 | 64.5 | 62.0 | 47.9 | 95.2 | 88.2 | 89.7 | 92.1 | 82.8 | 83.5 | 54.7 | 68.8 | 43.4 |
| | GWMAC | 4.4 | 15.4 | 0.4 | 30.6 | 27.2 | 12.2 | 11.4 | 0.2 | 0.1 | 16.1 | 4.0 | 1.8 | 3.2 | 2.2 | 0.2 |
| | UPMGC-SM | 53.9 | 76.3 | 36.6 | **76.1** | **71.3** | **61.9** | 61.9 | 55.4 | 44.5 | 64.8 | 59.7 | 46.3 | – | – | – |
| | SMILE | 50.9 | 79.4 | 35.2 | 71.1 | 70.4 | 58.2 | 97.9 | 94.2 | 95.4 | 98.6 | 96.3 | 97.0 | 57.8 | 77.1 | 48.8 |
| | CPMN | 59.2 | 76.9 | 69.3 | 68.5 | 71.1 | 58.4 | 95.8 | 90.2 | 91.5 | 96.6 | 92.0 | 93.1 | 67.2 | 80.9 | 53.5 |
| 50% Missing (ψ = 50%) | DAIMC | 56.2 | 78.0 | 41.8 | 62.7 | 58.5 | 47.7 | 33.8 | 26.4 | 16.0 | 55.2 | 49.6 | 38.6 | – | – | – |
| | EERIMVC | 43.6 | 69.0 | 26.4 | 68.7 | 63.9 | 53.8 | 55.6 | 45.9 | 36.8 | 65.2 | 55.7 | 48.9 | – | – | – |
| | SURE | 34.6 | 57.8 | 19.9 | 58.3 | 50.4 | 37.4 | 93.0 | 85.4 | 85.9 | 92.3 | 85.0 | 84.3 | 45.2 | 46.9 | 29.6 |
| | DCP | 44.3 | 71.0 | 45.3 | 53.7 | 65.5 | 47.3 | 80.0 | 75.2 | 70.7 | 94.0 | 89.7 | 88.3 | 26.3 | 47.2 | 14.4 |
| | DSIMVC | 16.4 | 24.8 | 9.2 | 54.4 | 52.4 | 35.2 | 55.8 | 55.1 | 43.0 | 97.0 | 92.4 | 93.5 | 29.4 | 48.5 | 19.0 |
| | SMILE | 51.2 | 79.0 | 35.6 | 69.5 | 66.7 | 54.9 | 96.8 | 91.7 | 93.0 | 98.5 | 95.7 | 96.6 | 54.6 | 76.3 | 45.2 |
| | CPMN | 52.9 | 79.7 | 51.7 | 67.8 | 60.4 | 54.5 | 96.0 | 93.3 | 91.2 | 97.5 | 95.8 | 97.0 | 58.1 | 75.0 | 50.3 |

birds. Following [41], visual features extracted by GoogLeNet [23] and text features extracted by doc2vec [14] are used as two views. (3) **NoisyMNIST** includes 70,000 instances, each providing two views: the raw MNIST image and a rotated, Gaussian-noised variant. Following [41], 30, 000 instances are randomly selected for evaluation. (4) **MNIST-USPS** contains 67,291 digit images from the MNIST and USPS datasets. As in a prior study [41], we randomly sample 5,000 items from each dataset, covering all 10 digits. (5) **YouTubeFaces** features 152,549 faces across 66 identities, with each identity having over 1,500 images. For comparisons, we follow prior work [41] to describe each image using multi-view features, specifically a 512-dimensional GIST feature, a 1984-dimensional HOG feature, and a 1024-dimensional HIST feature.

**Baselines** We compared CPMN with 12 multi-view clustering baselines. These baselines can be divided into the following categories based on the type of data they handle. Four baselines are designed to handle partially missing multi-view data, including DAIMC [9], EERIMVC [19], DCP [15], and DSIMVC [24]. PVC [10] and MvCLN [36] are designed for partially unaligned data. SURE [35] can address both of the aforementioned partial incompleteness problems. Distinctly, MVC-UM [39], GWMAC [5], and UPMGC-SM [32] are designed to handle fully unaligned multi-view data. DM2C [11] is designed to handle fully missing multi-view data. SMILE [41] can address both of the aforementioned problems. For fairness, we only show the clustering results of various methods on the incomplete data they can process.

**Implementation Details** We implement the proposed CPMN via the PyTorch public toolboxes, and we conduct the experiment on a server running Ubuntu Linux 20.04 that is equipped with an Intel(R) Xeon(TM) W5-3425 CPU @ 3.20 GHz, NVIDIA RTX 4090 Graphics Processing Units (GPUs), and 128 GB of memory. Following previous studies [35, 41], we define the missing rate as $\psi = m/N$, where $m$ is the number of instances with missing views. To generate incomplete data, we randomly select $m$ instances and drop one view for each instance. Similarly, we define the unaligned rate as $\zeta = c/N$, where $c$ is the number of instances without correspondences. We randomly sample $c$ instances and remove the correspondence between their views.

Table 3: ACC comparison with SMILE on Caltech-101 across a spectrum of missing view rates.

| Missing Rates | 20% | 40% | 60% | 80% | 100% |
|---|---|---|---|---|---|
| SMILE | **60.1** | **55.0** | 47.2 | 39.9 | 30.5 |
| CPMN | 59.4 | 54.8 | **48.4** | **42.3** | **38.7** |

Table 4: Ablation study on Caltech-101. Each configuration uses the loss terms identified by "✓".

| $L_{REC}$ | $L_{GL}$ | $L_{PL}$ | $L_{CLU}$ | 100% Missing | | | 100% Unaligned | | |
|---|---|---|---|---|---|---|---|---|---|
| | | | | ACC | ARI | NMI | ACC | ARI | NMI |
| ✓ | | | | 31.3 | 52.0 | 24.2 | 47.3 | 59.4 | 45.9 |
| ✓ | ✓ | | | 32.8 | 54.1 | 26.3 | 49.2 | 67.6 | 49.3 |
| ✓ | ✓ | ✓ | | 33.7 | 61.4 | 39.1 | 53.8 | 75.3 | 59.1 |
| ✓ | ✓ | | ✓ | 34.8 | 65.5 | 36.3 | 53.2 | 76.6 | 55.9 |
| ✓ | ✓ | ✓ | ✓ | **38.7** | **68.2** | **40.7** | **57.0** | **77.0** | **68.4** |

## 4.2 Comparison with Baselines

Table 1 provides a comprehensive quantitative comparison of CPMN against several multi-view clustering methods under two challenging fully incomplete scenarios: 100% unaligned instances and 100% missing views. The experimental results lead to several key observations. Firstly, our CPMN outperforms baselines across nearly all datasets, a success attributed to its novel prototype matching-based view alignment strategy. Secondly, with the unaligned rate of 100%, both CPMN and SMILE generally exhibit superior performance over other unaligned multi-view clustering methods, benefiting from the latent feature extraction capabilities of DNN. Conventional methods like UPMGC-SM achieve competitive performance, suggesting potential limitations of DNN-based approaches in capturing essential data distributions within smaller-scale datasets.

Furthermore, to comprehensively evaluate the robustness of CPMN, its clustering performance is also tested under partially incomplete scenarios. Table 2 presents the clustering results for CPMN alongside several baseline methods under scenarios of 50% unaligned instances and 50% missing views. The results indicate that even though CPMN utilizes no view correspondence information and employs view-specific autoencoders without weight sharing, it still achieves top-two performance across most datasets. Notably, it attains the best overall results on Caltech-101 and YouTubeFaces, leading in at least two metrics per dataset. These results further demonstrate the effectiveness of CPMN and its capability to generalize to partially incomplete settings.

To further evaluate the robustness of CPMN against the strongest baseline, SMILE, we conducted a detailed analysis on the Caltech-101 dataset with different missing view rates. Table 3 presents the clustering results under missing view rates from 20% to 100%. From the results, we can observe that: (1) At lower missing rates (20%-40%), where substantial paired information is still available, SMILE shows a slight advantage. Notably, our CPMN achieves highly competitive performance even in this scenario, for which it was not specialized, underscoring the general effectiveness of its correspondence-free framework. (2) Crucially, as the missing rate reaches 60% and paired information becomes scarce, SMILE's performance begins to degrade sharply. In contrast, CPMN's performance remains remarkably stable, surpassing SMILE. This trend intensifies as the problem difficulty increases (80%-100%), clearly demonstrating the superior robustness of CPMN's matching paradigm precisely in the challenging scenarios.

## 4.3 Ablation Studies and Parameter Analysis

To verify the significance of each component in CPMN, we perform ablation studies to isolate the effect of the reconstruction loss $L_{REC}$, graph contrastive loss $L_{GL}$, prototype contrastive loss $L_{PL}$, and clustering loss $L_{CLU}$. The clustering results on the Caltech-101 dataset with 100% unalignment and 100% missing views are shown in Table 4. From the results, one could observe that: (1) With the addition of components, the clustering performance improves significantly, and the best clustering performance can be achieved when using all loss terms. (2) $L_{REC}$ plays an essential role in the AEs

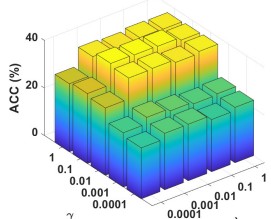 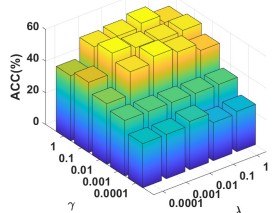 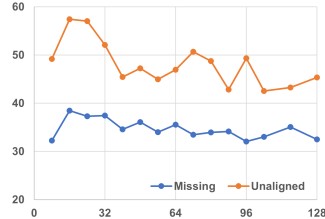

(a) ACC vs. $\gamma$ and $\lambda$ with fully missing multi-view data

(b) ACC vs. $\gamma$ and $\lambda$ with fully unaligned multi-view data

(c) ACC vs. $d_z$ with fully incomplete multi-view data

Figure 3: Sensitivity studies of CPMN on the hyper-parameter $\lambda$, $\gamma$, and $d_z$

and is therefore retained. Augmenting this with $L_{GL}$ led to a notable enhancement. (3) Building upon the $L_{REC} + L_{GL}$ baseline, the introduction of either $L_{PL}$ and $L_{CLU}$ can further improve performance. This underscores the complementary benefits of these terms in the overall model.

There are two tunable hyper-parameters, namely, $\lambda$ and $\gamma$, which impact the weights of various loss terms. To better illustrate the stability of CPMN, we conducted experiments to investigate the sensitivity of the proposed method to these two hyper-parameters. Figure 3a-3b shows the ACC values versus the hyper-parameters $\lambda$ and $\gamma$ on the Caltech-101 dataset with 100% unalignment and 100% missing views. The results show that the CPMN maintains relatively stable performance across a considerable range of these hyper-parameter settings within both incomplete scenarios. When $\lambda$ and $\gamma$ are selected from the ranges of [1e-3, 1] and [1e-2, 1], CPMN can achieve desirable performance.

The dimensionality of the latent space $d_z$ is a critical hyper-parameter in deep clustering models, exerting a considerable influence on clustering performance. To examine the effect of $d_z$, a sensitivity analysis is conducted on the Caltech-101 dataset with $d_z \in [8, 128]$. Figure 3c shows the ACC values versus $d_z$ with 100% unalignment and 100% missing views. It can be observed that ACC increases significantly as $d_z$ is raised from 8 to 16. Subsequently, clustering performance gradually declines with further increases in $d_z$. This phenomenon occurs because higher dimensionalities may introduce noise that adversely affects clustering, whereas lower dimensionalities might fail to capture the underlying structure.

## 5 Conclusion

In this paper, we propose the Contrastive Prototype Matching Network (CPMN), a novel framework that establishes a new structure-level alignment paradigm for fully incomplete multi-view clustering (FIMC). CPMN pioneers learning consistent global clustering by explicitly aligning prototypes across views. Initially, a correspondence-free graph contrastive learning stage with MNN establishes view-specific prototypes from entirely unpaired views. Subsequently, a crucial similarity-based prototype graph matching step explicitly aligns these prototypes, resolving category discrepancies and forging a unified clustering structure across views. Building on this alignment, CPMN employs a prototype-aware contrastive learning mechanism that leverages matched prototypes and emerging category information to refine representations for enhanced semantic consistency. Extensive experiments demonstrate that CPMN achieves superior performance compared to various baselines.

Despite its success, we acknowledge avenues for future work. The performance of CPMN is theoretically linked to the quality of the initial prototypes, suggesting further research into the convergence properties of the joint optimization. Furthermore, our method assumes a predefined cluster count $C$. Extending this structural alignment paradigm to a non-parametric setting, where $C$ is dynamically inferred from the data, remains a significant and open research direction.

## Acknowledgments

This work was supported in part by the National Natural Science Foundation of China under Grant 62402235, 62276143 and 62272035, and the Natural Science Research Start-up Foundation of Recruiting Talents of Nanjing University of Posts and Telecommunications under Grant NY223165.

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
