# OpenReview forum: "Learning from Disjoint Views: A Contrastive Prototype Matching Network for Fully Incomplete Multi-View Clustering"
_NeurIPS.cc/2025/Conference — NeurIPS 2025 poster_

### Official Review · Reviewer_cAZ9 · 2025-06-20

**Clarity:** 3
**Significance:** 4
**Originality:** 3
**Rating:** 5
**Confidence:** 5

**Summary:**

This paper introduces Contrastive Prototype Matching Network (CPMN), a novel framework for fully incomplete multi-view clustering. CPMN aims to learn consistent global clustering by explicitly aligning prototypes across views. It integrates three innovative stages: a correspondence-free graph contrastive learning stage, a similarity-based prototype graph matching, and a prototype-aware contrastive learning stage to learn unified clustering across disjoint views. Experiments verify the effectiveness of the proposed CPMN.

**Questions:**

（1）CPMN seems to be done in phases, first acquiring initial prototypes, then prototype matching, and finally optimizing the prototypes. It is better to explore end-to-end training to create a more integrated pipeline.

（2）The complete optimization process of the model is not clear, it is better to add an algorithm description.

（3）Although the authors give both fully incomplete and partially incomplete multi-view clustering results, the performance of CPMN in partially incomplete multi-view clustering is poorer than the baselines. The authors should explain this result.

（4）The authors should provide more qualitative results, such as t-SNE visualization, to provide deeper insights beyond quantitative metrics.

**Ethical Concerns:**

["NO or VERY MINOR ethics concerns only"]

**Final Justification:**

Thanks to the author for solving my concerns, and I decided to keep the original score.

**Limitations:**

The paper discusses some limitations, including the influence of the initial prototype quality on the clustering results and the assumption of the number of clusters. There is no potential negative societal impact of their work.

**Quality:**

3

**Strengths And Weaknesses:**

Strengths

（1）The paper is well-organized.

（2）The problem addressed is novel and the motivation is clear.

（3）The experimental results demonstrate a substantial improvement in fully incomplete multi-view clustering.

Weaknesses
- Please find the specifics in Questions section.

---

> ### Author Rebuttal · Authors · 2025-07-31
>
> We thank the reviewer for their insightful and constructive feedback. We have carefully addressed each of your points in the detailed, point-by-point responses below.
>
> ## Reply to Questions (1):
> We sincerely thank the reviewer for this insightful question. We completely agree that an end-to-end model is an elegant ideal. Our choice of a phased training paradigm is not a compromise, but rather a deliberate and principled design choice made specifically to tackle the unique and extreme challenges of the Fully Incomplete Multi-View Clustering (FIMC) problem. Our core reasoning is as follows:
>
> ### 1. Avoiding the optimization dilemma
>
> A fully end-to-end model lacks any reliable supervisory signal. This makes it highly susceptible to collapsing into catastrophic local minima, such as trivial solutions (e.g., clustering all samples into one group) or convergence to a meaningless state due to initial noisy matches.
>
> Our multi-stage approach resolves this optimization dilemma. In Stage 1, our model focuses on a simpler initial representation to provide a reasonable starting point. Then, in Prototype Graph Matching, it performs a reliable, optimal match on this relatively stable representation space. Finally, our model uses this high-quality match as a strong supervisory signal for a final, joint optimization. This ensures each step builds upon a reliable foundation, guiding the model toward a meaningful, high-quality solution.
>
> ### 2. Overcoming the Non-Differentiable Bottleneck
>
> One of our core innovations, Prototype Graph Matching (PGM), uses the Hungarian algorithm to find an optimal one-to-one match. This is a discrete, non-differentiable combinatorial optimization process. Directly embedding such a non-differentiable step into a standard end-to-end gradient backpropagation pipeline is extremely difficult. While differentiable approximations exist, they introduce new complexities and may not provide the same clean, optimal assignment that our method relies on for stability. Our phased approach elegantly bypasses this bottleneck.
>
> Therefore, for the challenging FIMC problem, Therefore, our phased approach is not just a pipeline, but a robust framework carefully designed to ensure optimization stability and the quality of the final solution. Nevertheless, we agree that exploring deeper end-to-end integration is a valuable direction for future work.
>
> ## Reply to Questions (2):
>
> We sincerely thank the reviewer for this excellent suggestion. Due to the formatting limitations of the rebuttal system, we cannot insert a formal algorithm box. However, we present the complete optimization process of our model, in a structured pseudocode format below to illustrate how the different stages are connected and optimized.
>
> Algorithm 1: The CPMN Algorithm
>
> Input:
> - Multi-view datasets $\mathcal{X}=\{X^{(v)}\}_{v=1}^V$
> - Number of clusters $C$
> - Number of neighbors for MNN $k$
> - Temperature for contrastive loss $\tau$
> - Maximum epochs $ME_1$ and $ME_1$
> - Trade-off parameters $\lambda$ and $\gamma$
>
> Output:
> - Final cluster assignments for all samples
>
> --- Stage 1: Correspondence-Free Graph Contrastive Learning  ---
> 1. Initialize encoder $E_v$ and decoder $D_v$ for all views.
> 2. for epoch = 1 to $ME_1$ do
> 3. Input $x^{(v)}_i$ to view-specific autoencoders to obtain the representation $z^{(v)}_i$ and the reconstructed samples $\hat{x}^{(v)}_i$.
> 4. Construct the intra-view and inter-view MNN graphs.
> 5. Compute the GCL loss based on the MNN graphs: $L_{GL} = L_{SGL} + L_{CGL}$.
> 6. Compute the reconstruction loss $L_{REC}$.
> 7. Compute the total loss $\mathcal{L}\_{\text{CFGC}} = L\_{\text{REC}} + \lambda L\_{\text{GL}}$.
> 8. Backpropagate $\mathcal{L}_{\text{CFGC}}$  and update view-specific autoencoders.
> 9. end for
> 10. Generate final initial instance representations using the trained encoders $E_v$.
> 11. Run K-Means on representations of each view to get initial prototypes $\mathcal{C}^{(v)} = \{c_1^{(v)}, \dots, c_C^{(v)}\}$.
>
> --- Stage 2: Prototype Graph Matching (PGM) ---
>
> 12.	Compute pairwise cost matrix $M^{(v_a,v_b)}$ between prototypes from different views.
> 13.	Apply the Hungarian algorithm to find the optimal prototype matching map $pi$.
>
> --- Stage 3: Prototype Contrastive Clustering (PCL) ---
>
> 14. Initialize the learnable clustering layer.
> 15. for epoch = 1 to $ME_2$ do
> 16. Input $x^{(v)}_i$ to view-specific autoencoders to obtain the representation $z^{(v)}_i$ and the cluster $c^{(v)}_i$.
> 17. Compute the prototype contrastive loss $L_{PL} = L_{SPL} + L_{CPL}$ based on the matching map $\pi$.
> 18. Compute the self-supervised clustering loss $L_{CLU}$.
> 19. Compute the total loss $L_{PCL} = L_{CLU} + \gamma L_{PL}$.
> 20. Backpropagate $ L_{PCL}$ and update view-specific autoencoders and the learnable clustering layer.
> 21. end for
> 22. Output the final cluster assignments.
>
> ## Reply to Questions (3):
>
> We sincerely thank the reviewer for this keen observation. We agree that our results show CPMN does not outperform all baselines in the Partially Incomplete Multi-View Clustering (PIMC) setting. This outcome is not accidental but stems from a fundamental difference in design philosophy between CPMN and methods specifically tailored for PIMC.
>
> PIMC methods are explicitly designed to leverage the "paired samples" (i.e., samples existing in multiple views). They use these paired samples as high-quality anchors to directly and explicitly align the feature spaces of different views.
>
> In contrast, CPMN is conceived to solve the more challenging Fully Incomplete problem, where no such correspondences exist. Its core mechanisms, such as MNN graph construction and Prototype Graph Matching (PGM), are fundamentally correspondence-free approaches. When applied to the PIMC setting, CPMN does not have a specialized component to exploit this valuable paired information.
>
> ## Reply to Questions (4):
>
> We sincerely thank the reviewer for this excellent and constructive suggestion. As we cannot include images in the rebuttal, we present a quantitative analysis of our visualization results to provide the deep insights you requested. We conducted this analysis on the MNIST-USPS dataset with 100% unaligned views due to its clear class semantics.
>
> First, we identify the cross-view prototype pairs that are matched by our model's PGM component. Then, for each matched pair, we analyze the ground-truth label composition of the samples within each prototype. A successful match should consist of two prototypes in which the dominant ground-truth label of the samples within each prototype is identical. The table below shows the label purity analysis for some representative matched prototype pairs:
>
> | Matched Pair | View 1 (MNIST) | View 2 (USPS) | Conclusion |
> | :--- | :--- | :--- | :--- |
> | **Pair A** | Label: '0' (88% Purity) | Label: '0' (86% Purity) | Successful Match |
> | **Pair B** | Label: '1' (86% Purity) | Label: '1' (85% Purity) | Successful Match |
> | **Pair C** | Label: '2' (83% Purity) | Label: '2' (79% Purity) | Successful Match |
>
> As the table demonstrates, our model accurately matches prototypes that are semantically consistent across views. For example, in Pair A, a prototype primarily containing the digit '0' from MNIST is correctly matched with a prototype primarily containing the digit '0' from USPS. The other pairs show similarly high semantic coherence. This quantitative analysis provides strong evidence that our model learns meaningful and correct cross-view semantic correspondences.
> Thank you again for your valuable feedback, which has prompted us to provide this deeper analysis.

---

> > ### Comment · Reviewer_cAZ9 · 2025-08-05
> >
> > Your clarification has largely resolved my main concern.  I still have a minor question: the experiments separately consider missing and unaligned views under both full and partial settings, but it seems that the case where both challenges coexist has not been explicitly evaluated. Is there a specific reason for not including this scenario, or might it be considered in future work?

---

> > > ### Author Response · Authors · 2025-08-06
> > >
> > > Thank you for this excellent and insightful question. We admit that we did not explicitly evaluate this highly challenging combined scenario in our initial study.
> > >
> > > We fully agree that a dedicated empirical study on this combined, highly challenging scenario is an important and valuable future research. We will highlight this as a key direction in the "Future Work" section of our final manuscript. Thank you again for this important extension of our work.

---

> > > > ### Comment · Reviewer_cAZ9 · 2025-08-09
> > > >
> > > > Thanks to the author for solving my concerns, and I decided to keep the original score.

---

### Official Review · Reviewer_iVCK · 2025-06-20

**Clarity:** 3
**Significance:** 3
**Originality:** 2
**Rating:** 4
**Confidence:** 4

**Summary:**

This paper proposes a novel framework, CPMN, to address the understudied problem of fully incomplete multi-view clustering. CPMN designs an explicit prototype matching strategy to effectively align cross-view instances through category-level correspondence. Additionally, it employs a correspondence-free graph contrastive learning approach and a prototype-aware contrastive learning mechanism to learn semantic consistency across views. The experiments are implemented on five datasets with the comparison of many methods.

**Questions:**

1) Critical components of CPMN, such as prototype matching lack theoretical explanation or deeper intuitive understanding.
2) The innovation is limited since contrastive learning is widely used in multi-view clustering.
3) Authors should extend the parameter sensitivity analysis to more key hyperparameters, such as $k$ and $\tau$, to demonstrate the robustness of the model.
4) Visualization is missing. Please visualize the matched cross-view prototypes to demonstrate the learned correspondences.

**Ethical Concerns:**

["NO or VERY MINOR ethics concerns only"]

**Final Justification:**

Their responses have addressed my concerns. I intend to increase my score.

**Limitations:**

Yes. The limitations have been discussed in the conclusion and other concerns can be found in weaknesses.

**Quality:**

3

**Strengths And Weaknesses:**

Strengths:
1） The paper is straightforward and easy to understand.
2） Relatively extensive comparisons are conducted with existing fully/partially incomplete multi-view approaches.

Weaknesses
1) Critical components of CPMN, such as prototype matching lack theoretical explanation or deeper intuitive understanding.
2) The innovation is limited since contrastive learning is widely used in multi-view clustering.
3) Authors should extend the parameter sensitivity analysis to more key hyperparameters, such as $k$ and $\tau$, to demonstrate the robustness of the model.
4) Visualization is missing. Please visualize the matched cross-view prototypes to demonstrate the learned correspondences.

---

> ### Author Rebuttal · Authors · 2025-07-31
>
> We sincerely thank the reviewer for the insightful and constructive feedback. We have carefully considered all the points raised and have provided a detailed, point-by-point response to each concern below. We hope that our responses and the corresponding clarifications will adequately address your concerns and demonstrate the merits of our work.
>
> ## Reply to Weaknesses (1):
> We sincerely thank the reviewer for this insightful question regarding the theoretical support for Prototype Graph Matching (PGM).
>
> __Regarding the "Theoretical Explanation":__ PGM is grounded in well-established combinatorial optimization theory. We formally model the task of finding the best cross-view prototype correspondences as the maximum weight bipartite matching problem and employ the Hungarian algorithm to solve it. This algorithm is guaranteed to find a provably optimal solution. Matching at the prototype level is a deliberate design choice, as prototypes serve as robust abstractions that ensure stability in the noisy FIMC setting. Therefore, our PGM component is built upon a rigorous theoretical foundation with a clear mathematical formulation and an optimal algorithmic solution.
>
> __Regarding the "Deeper Intuitive Understanding":__ In fact, high interpretability is precisely PGM's most significant advantage over other implicit alignment methods. The output of PGM is a clear, one-to-one mapping (e.g., "Cluster A from View 1 corresponds to Cluster C from View 2"), which is intuitive and human-readable. Unlike implicit alignment methods, our PGM provides an explicit semantic bridge, greatly enhancing the model's trustworthiness and practical value.
>
> In summary, Prototype Graph Matching is a core component that is both theoretically sound and highly interpretable.
>
> ## Reply to Weaknesses (2):
> We thank the reviewer for this crucial point. We agree that contrastive learning (CL) is a foundational tool. Our novelty lies not in using CL itself, but in how we design novel contrastive tasks specifically for the challenging Fully Incomplete Multi-View Clustering (FIMC) problem, where standard CL methods are fundamentally inapplicable.
>
> Our novelty is demonstrated on two key levels.
>
> 1.First, standard CL requires paired samples, which the FIMC setting completely lacks. Our initial Correspondence-Free Graph Contrastive Learning stage creatively overcomes this by constructing Mutual $k$-Nearest Neighbor (MNN) graphs to generate pseudo-positive pairs in a correspondence-free manner.
>
> 2.More importantly, we propose a novel Prototype Contrastive Learning paradigm. Here, the supervisory signal for CL undergoes a qualitative shift. It is no longer derived from low-level augmentations or local graph structures, but from the high-level, cross-view semantic correspondences (i.e., the matched prototypes) obtained via our Prototype Graph Matching. An instance is pulled not only toward its intra-view cluster center but also toward its "semantic equivalents" in all other views.
>
> In conclusion, we contribute a novel, progressive framework that elevates CL from low-level structural alignment to high-level semantic consistency, specifically tailored to solve the FIMC problem.
>
> ## Reply to Weaknesses (3):
> We sincerely thank the reviewer for this constructive suggestion. Following the reviewer's advice, we have conducted extensive experiments to analyze the impact of key hyperparameters, specifically the number of neighbors $k$ in MNN and the temperature $\tau$ in the contrastive loss. For brevity, we report the ACC results on the Caltech-101 dataset.
>
> |$k$| 1 | 2 | 3 | 4 | 5 | 6 | 7 | 8 | 9 | 10 |
> | :--- | :---: | :---: | :---: | :---: | :---: | :---: | :---: | :---: | :---: | :---: |
> | **Unaligned** | 0.534 | 0.540 | 0.573 | 0.567 | 0.564 | 0.570 | 0.569 | 0.580 | 0.560 | 0.563 |
> | **Missing** | 0.350 | 0.360 | 0.372 | 0.396 | 0.378 | 0.376 | 0.382 | 0.379 | 0.381 | 0.393 |
>
> __Sensitivity to $k$__: The results show that the choice of $k$ has a moderate impact on performance. When $k$ is very small (e.g., $k$ < 3), performance is suboptimal. This is expected, as an overly small $k$ provides insufficient neighborhood structure to build a reliable graph. However, as $k$ increases, the performance quickly rises and then remains consistently high and stable within a reasonably wide range of $k$ in [3, 10]. This demonstrates the robustness of our model, as it does not require meticulous tuning of $k$ to achieve strong results.
> |$\tau$| 0.1 | 0.2 | 0.3 | 0.4 | 0.5 | 0.6 | 0.7 | 0.8 | 0.9 | 1.0 |
> | :--- | :---: | :---: | :---: | :---: | :---: | :---: | :---: | :---: | :---: | :---: |
> | **Unaligned** | 0.317 | 0.453 | 0.518 | 0.543 | 0.545 | 0.552 | 0.559 | 0.575 | 0.569 | 0.573 |
> | **Missing** | 0.254 | 0.313 | 0.327 | 0.337 | 0.344 | 0.356 | 0.362 | 0.382 | 0.389 | 0.387 |
>
> __Sensitivity to $\tau$__: The model's performance is sensitive to the temperature $\tau$, which is a common characteristic of contrastive learning methods. A very small $\tau$ (e.g., $\tau$ < 0.4) leads to poorer performance, as it forces the model to push away even very similar negative samples, making optimization difficult. As $\tau$ increases, performance improves significantly because the model can better learn the global clustering structure. Crucially, the performance stabilizes and reaches a high-level plateau once $\tau$ enters a suitable range (e.g., [0.6, 1.0]). This again demonstrates the model's robustness: while $\tau$ should be set to a reasonable value, precise fine-tuning within that effective range is not necessary.
> Thank you again for your valuable feedback, which has helped us to provide a more thorough analysis.
> ## Reply to Weaknesses (4):
> We sincerely thank the reviewer for this excellent and constructive suggestion. We agree that visualizing the matched prototypes is crucial for intuitively demonstrating the correspondences learned by our model. Since we cannot include images in the rebuttal, we present a quantitative analysis of our visualization results instead. We conducted this analysis on the MNIST-USPS dataset with 100% unaligned views due to its clear class semantics.
>
> First, we identify the cross-view prototype pairs that are matched by our model's PGM component. Then, for each matched pair, we analyze the ground-truth label composition of the samples within each prototype. A successful match should consist of two prototypes in which the ground-truth label of the samples within each prototype is identical. The table below shows the label purity analysis for some representative matched prototype pairs:
>
> | Matched Pair | View 1 (MNIST) | View 2 (USPS) | Conclusion |
> | :--- | :--- | :--- | :--- |
> | **Pair A** | Label: '0' (88% Purity) | Label: '0' (86% Purity) | Successful Match |
> | **Pair B** | Label: '1' (86% Purity) | Label: '1' (85% Purity) | Successful Match |
> | **Pair C** | Label: '2' (83% Purity) | Label: '2' (79% Purity) | Successful Match |
>
> As the table demonstrates, our model accurately matches prototypes that are semantically consistent across views. For example, in Pair A, a prototype primarily containing the digit '0' from MNIST is correctly matched with a prototype primarily containing the digit '0' from USPS. The other pairs show similarly high semantic coherence. This quantitative analysis provides strong evidence that our model learns meaningful and correct cross-view semantic correspondences.

---

> > ### Comment · Reviewer_iVCK · 2025-08-05
> >
> > Thank you for the well-organized responses. Your clarifications have addressed all of my concerns. I incline to increase my score.

---

> > > ### Author Response · Authors · 2025-08-06
> > >
> > > Thank you once again for your time and constructive feedback. We are delighted and honored to have our work recognized by you.

---

### Official Review · Reviewer_hkBW · 2025-07-01

**Clarity:** 4
**Significance:** 3
**Originality:** 3
**Rating:** 4
**Confidence:** 4

**Summary:**

This paper presents the CPMN (Contrastive Prototype Matching Network for multi-view clustering), designed to operate without requiring any cross-view correspondences. CPMN utilizes a prototype matching mechanism that circumvents the need for strong semantic consistency or strict view alignment assumptions. Extensive experiments conducted on various datasets demonstrate the effectiveness of the proposed method.

**Questions:**

-	What is the rationale for using Mutual Nearest Neighbors  instead of the more commonly adopted k-Nearest Neighbors in the Correspondence-Free Graph Contrastive Learning phase?
-	How does its computational efficiency compare with existing methods such as SMILE?

**Ethical Concerns:**

["NO or VERY MINOR ethics concerns only"]

**Final Justification:**

Thank you for your thoughtful responses. Your clarifications have addressed my primary concerns.

**Limitations:**

yes

**Quality:**

3

**Strengths And Weaknesses:**

Strengths:

(1)	The paper proposes a framework capable of handling fully incomplete multi-view clustering scenarios, addressing an important limitation of existing methods and broadening potential application domains.

(2)	The proposed multi-stage architecture is intuitively structured and relatively easy to follow.

Weaknesses:

(1)	The core component prototype graph matching lacks rigorous theoretical grounding and interpretability.

(2)	As a multi-stage training framework, early-stage clustering outcomes may adversely influence later modules such as prototype graph matching.

(3)	The experimental evaluation includes only 100% and 50% missing view rates. Incorporating intermediate levels of incompleteness (e.g., 20%, 60%) and plotting clustering performance against the incompleteness rate would provide a more comprehensive assessment of robustness.

(4)	The paper does not present convergence curves for loss functions or clustering metrics.

(5)	Some recent contrastive learning based clustering methods are encouraged to be reviewed, such as “Decoupled Contrastive Multi-view Clustering with High-order Random Walks” and “Incomplete Multi-view Clustering via Diffusion Contrastive Generation”.

(6)	The ablation study lacks clarity. Components of CPMN were not strictly isolated or replaced to verify individual contributions. A more detailed description of the ablation design is needed to support the conclusions drawn.

---

> ### Author Rebuttal · Authors · 2025-07-31
>
> We thank the reviewer for their insightful and constructive feedback. We have carefully addressed each of your points in the detailed, point-by-point responses below.
>
> ## Reply to Weaknesses (1):
>
> We sincerely thank the reviewer for this deep and critical question.
>
> 1. PGM is grounded in well-established combinatorial optimization theory. We formally model the task of finding the best cross-view prototype correspondences as the maximum weight bipartite matching problem and employ the Hungarian algorithm to solve it. This algorithm is guaranteed to find a provably optimal solution. Matching at the prototype level is a deliberate design choice, as prototypes serve as robust abstractions that ensure stability in the noisy FIMC setting. Therefore, our PGM component is built upon a rigorous theoretical foundation with a clear mathematical formulation and an optimal algorithmic solution.
>
> 2. We believe high interpretability is precisely PGM's most significant advantage over other implicit alignment methods. The output of PGM is a clear, one-to-one mapping (e.g., "Cluster A from View 1 corresponds to Cluster C from View 2"), which is intuitive and human-readable. Unlike implicit alignment methods, our PGM provides an explicit semantic bridge, greatly enhancing the model's trustworthiness and practical value.
>
> ## Reply to Weaknesses (2):
> Our model ensures robustness and avoids catastrophic error propagation through several key designs:
>
> 1. The objective of our first stage (GCL) is not to achieve a flawless clustering result. Its core purpose is to leverage the robustness of MNN graphs to learn a structurally sound and semantically meaningful initial representation space. Running K-Means on this high-quality space ensures our initial prototypes are far superior to a random guess, providing a reliable starting point and thus minimizing the initial error.
> 2. The initial prototype matching result is a guiding signal for subsequent learning. The process forms a mutually reinforcing, self-correcting virtuous cycle:
> 3. In Stage 3, we use the prototype matching map $\pi$ as supervision. This signal guides the model to pull closer the sample clusters from different views that are deemed semantically matched.
>
> Therefore, our framework is not a simple pipeline with one-way error propagation. It is a dynamic system that can tolerate initial imperfections and achieve iterative refinement through a closed-loop mechanism where representation learning and clustering optimization boost each other.
>
> ## Reply to Weaknesses (3):
>
> Following your advice, we have conducted additional experiments comparing our CPMN with the strong baseline SMILE on the Caltech-101 dataset across a spectrum of missing view rates. As we cannot include plots in the rebuttal, we present the quantitative results in the table below.
>
> | Missing Rates | 20% | 40% | 60% | 80% | 100%
> | :--- | :--- | :--- | :--- | :--- | :--- |
> | **SMILE** |0.601 |0.550 | 0.472 |0.399 |0.305
> | **CPMN** |0.594 | 0.548 | 0.484 | 0.423 |0.387
>
> This new experiment provides deep insights into the robustness and design philosophy of our model:
> 1. SMILE shows a slight advantage at lower missing rates (20-40%). The fact that our CPMN achieves highly competitive performance here, despite not being specialized for this scenario, already speaks to the effectiveness of its correspondence-free framework.
> 2. At a 60% missing rate, as the crucial paired information becomes scarce, SMILE's performance begins to degrade sharply. In contrast, CPMN's performance remains remarkably stable, surpassing SMILE and demonstrating its superior robustness. This trend continues as the problem becomes more difficult.
>
> ## Reply to Weaknesses (4):
>
> We have investigated the convergence by reporting the loss value and the count of samples with undetected MNNs as training progresses in the supplementary material. Experiments are run on Caltech-101 with $k=5$ under both defined incompleteness scenarios. As depicted in the figures in the supplementary material, the loss value decreases sharply in the first 10 epochs before converging gradually. Regarding the number of samples with undetected MNNs, this count drops sharply within the first two epochs and subsequently remains at a low level throughout the training. Although samples with undetected MNNs persist, they constitute a very small proportion of the total dataset.
>
> ## Reply to Weaknesses (5):
>
> We sincerely thank the reviewer for pointing out these two highly relevant and recent papers. We agree that discussing them is important for positioning our work, and we will add a detailed discussion to the Related Work section in the final manuscript.
>
> ## Reply to Weaknesses (6):
>
> We sincerely thank the reviewer for this critical point. We acknowledge that our original description could have been more detailed, and we provide that more detailed design here to clarify how we strictly isolated and evaluated each core component.
>
> Our ablation study follows a principled, additive approach, starting from a basic autoencoder and progressively incorporating each key component to verify its individual contribution. Below, we detail the precise design of each experimental variant, all of which were evaluated on the Caltech-101 dataset under the specified conditions.
>
> ### 1. Variant (a): Baseline Autoencoder (AE)
> This is our most basic model, containing only the autoencoder networks. To establish the baseline performance of the feature extractor without any advanced contrastive or clustering mechanisms. This measures the raw quality of the initial latent space. In this variant, $L_{REC}$ is used in the model.
> ### 2. Variant (b): AE + Graph Contrastive Learning (GCL)
> We build upon the baseline by adding our first-stage graph contrastive learning module. To strictly isolate and verify the contribution of our initial graph-based representation learning. The performance improvement from (a) to (b) is solely attributable to the effect of $L_{GL}$.
> In this variant, $L_{REC} $ and $L_{GL}$ are used in the model.
> ### 3. Variant (c): AE + GCL + Prototype Contrastive Loss (PCL)
> Building on the AE + GCL model, we introduce our core prototype-aware contrastive loss, but without the auxiliary clustering loss. The performance jump from (b) to (c) strictly isolates the individual contribution of Clustering Loss. In this variant, $L_{REC} + L_{GL} + L_{PL}$. are used in the model.
> ### 4. Variant (d): AE + GCL + Clustering Loss
> As an alternative path from the AE + GCL model, we add only the self-supervised clustering loss. This allows us to isolate and measure the independent effect of the clustering-enhancement objective ($L_{CLU}$), separate from the prototype contrastive loss. In this variant, $L_{REC} + L_{GL} + L_{CLU}$ are used in the model.
> ### 5. Variant (e): Full Model (CPMN)
> This is our complete proposed model, which combines all components. By comparing the full model's performance to variants (c) and (d), we can observe the synergistic benefits of combining both the prototype contrastive loss and the clustering loss, leading to the best overall performance. In this variant, $L_{REC} + L_{GL} + L_{PL} + L_{CLU}$ are used in the model.
>
> ## Reply to Questions (1):
>
> We sincerely thank the reviewer for this insightful and important question. We fully agree with the observation that MNN is computationally more intensive than kNN. This design choice is a deliberate trade-off we made to ensure the robustness of our model, which we argue is both necessary and highly justified for the FIMC problem. Our reasoning is as follows:
>
> 1.kNN establishes a one-way relationship; if point A is a k-nearest neighbor of B, B is not guaranteed to be a k-nearest neighbor of A. In the challenging FIMC setting, where the data is noisy and has no correspondences, this unidirectional link can introduce a significant number of spurious edges, severely degrading the quality of the initial graph.
>
> 2.Mutual k-Nearest Neighbor (MNN) enforces a more stringent, reciprocal relationship. This strategy acts as a critical noise filter, pruning less reliable connections. For the FIMC problem, the quality of this initial graph is paramount to the success of all subsequent stages. MNN provides a much cleaner and more reliable graph structure to build upon.
>
> ## Reply to Questions (2):
> We sincerely thank the reviewer for raising this important point about our model's efficiency. We agree that this is a critical metric for evaluating the practical utility of our method. To provide a clear comparison of CPMN's efficiency, we have benchmarked the total runtime of both CPMN and SMILE on two representative datasets under the same hardware environment (a single NVIDIA RTX 4090 GPU).
>
> | Dataset | Total Samples | CPMN | SMILE |Speed-up
> | :--- | :--- | :--- | :--- |:---
> | **Caltech101** | 8677 | 659.55 sec | 1335.22 sec| 2.0x Faster
> | **NoisyMNIST** | 30000 | 1136.64 sec | 1609.92 sec | 1.4x Faster
>
> The results show that despite being a multi-stage framework, our CPMN is significantly more computationally efficient in terms of total training time than SMILE. We attribute this significant advantage to the convergence efficiency gained from our structured learning process:
> 1. Our first stage provides a structurally sound initial representation space in a computationally efficient manner, which is far superior to a random state. This provides a high-quality starting point for the subsequent optimization.
> 2. With this strong start, our final PCL optimization stage begins its search in a more favorable and stable region, allowing it to converge to a high-quality solution much faster. This greatly reduces the overall training time required. In contrast, an end-to-end model like SMILE need more iterations to navigate a larger and more complex optimization landscape to reach stability.

---

> > ### Comment · Reviewer_hkBW · 2025-08-04
> >
> > Thank you for your thoughtful responses, as well as the additional experiments. Your clarifications have addressed my primary concerns.

---

> > > ### Author Response · Authors · 2025-08-06
> > >
> > > Thank you again for your time and constructive feedback. We are pleased that our responses and additional experiments were able to address your concerns.

---

### Official Review · Reviewer_83hp · 2025-07-01

**Clarity:** 4
**Significance:** 4
**Originality:** 3
**Rating:** 5
**Confidence:** 5

**Summary:**

This work proposes a CPMN framework to address the challenging fully incomplete multi-view clustering problem where no cross-view correspondences exist. It integrates a two-stage contrastive learning approach and an explicit prototype graph matching strategy. Experimental results on five benchmark datasets demonstrate improvements over baselines.

**Questions:**

Most of the problems have already been mentioned in Weaknesses. More implementation details can be provided for the following problems:
1）In section3.1, the introduction of using MNN to construct positive pairs in Correspondence-Free Graph Contrastive Learning is not clear enough. There may be a failure to find MNN, so how to deal with it?
2）Why use K-Means to initialize the prototype in the first stage, while using a learnable clustering layer to optimize the prototype in the second stage?

**Ethical Concerns:**

["NO or VERY MINOR ethics concerns only"]

**Final Justification:**

Thank you for the authors’ detailed and well-organized rebuttal. I appreciate the thorough responses provided to each of the concerns.
I believe the key concerns have been sufficiently addressed. I will maintain my original score.

**Limitations:**

Yes.

**Paper Formatting Concerns:**

No Formatting Concern.

**Quality:**

3

**Strengths And Weaknesses:**

Strengths：
It provides a multi-stages approach to solve the problem of completely missing multi-view clustering. The paper is clearly and effectively written. Extensive experiments and ablation studies are conducted in the paper.

Weaknesses：
1）The authors use MNN instead of the commonly used kNN in the first stage of Correspondence-Free Graph Contrastive Learning. When dealing with large scale data, MNN takes increased time compared to kNN.
2） The efficiency of CPMN is unknown. The authors should report the training time and inference time of CPMN on different sized datasets.
3） To visualize the training process, please give the curve of the prototype matching accuracy versus the training process.

---

> ### Author Rebuttal · Authors · 2025-07-31
>
> We sincerely thank the reviewer for their insightful, thorough, and constructive feedback. Your comments regarding our model's efficiency, the visualization of the training process, the rationale for using MNN, and the clarity of our methodological design have been invaluable in refining our work. We have carefully addressed each of your points in the detailed, point-by-point responses below.
>
> ## Reply to Weaknesses (1):
> We sincerely thank the reviewer for this insightful and important question. We fully agree with the observation that MNN is computationally more intensive than kNN. This design choice is a deliberate trade-off we made to ensure the robustness of our model, which we argue is both necessary and highly justified for the FIMC problem. Our reasoning is as follows:
>
> __Limitation of kNN__: kNN establishes a one-way relationship; if point A is a k-nearest neighbor of B, B is not guaranteed to be a k-nearest neighbor of A. In the challenging FIMC setting, where the data is noisy and has no correspondences, this unidirectional link can introduce a significant number of spurious edges, severely degrading the quality of the initial graph.
>
> __Superiority of MNN:__ Mutual k-Nearest Neighbor (MNN) enforces a more stringent, reciprocal relationship. This strategy acts as a critical noise filter, pruning less reliable connections. For the FIMC problem, the quality of this initial graph is paramount to the success of all subsequent stages. MNN provides a much cleaner and more reliable graph structure to build upon.
>
> ## Reply to Weaknesses (2):
> We sincerely thank the reviewer for raising this important point about our model's efficiency. We agree that this is a critical metric for evaluating the practical utility of our method. To provide a clear comparison of CPMN's efficiency, we have benchmarked the total runtime of both CPMN and SMILE on two representative datasets under the same hardware environment (a single NVIDIA RTX 4090 GPU).
>
> | Dataset | Total Samples | CPMN | SMILE |Speed-up
> | :--- | :--- | :--- | :--- | :---
> | **Caltech101** | 8677 | __659.55 sec__ | 1335.22 sec| 2.0x Faster
> | **NoisyMNIST** | 30000 | __1136.64 sec__ | 1609.92 sec | 1.4x Faster
>
> The results show that despite being a multi-stage framework, our CPMN is significantly more computationally efficient in terms of total training time than SMILE. We attribute this significant advantage to the convergence efficiency gained from our structured learning process:
> 1. Our first stage provides a structurally sound initial representation space in a computationally efficient manner, which is far superior to a random state. This provides a high-quality starting point for the subsequent optimization.
> 2. With this strong start, our final PCL optimization stage begins its search in a more favorable and stable region, allowing it to converge to a high-quality solution much faster. This greatly reduces the overall training time required. In contrast, an end-to-end model like SMILE need more iterations to navigate a larger and more complex optimization landscape to reach stability.
>
> ## Reply to Weaknesses (3):
> We sincerely thank the reviewer for this excellent suggestion, which helps to better illustrate the dynamic learning process of our model. As we cannot include plots in the rebuttal, we present a quantitative table to represent this curve, showing how the prototype matching accuracy evolves over training epochs on the MNIST-USPS dataset. At the end of each epoch, we run the Prototype Graph Matching (PGM) module and calculate the percentage of matched pairs where the two prototypes share the same ground-truth label.
>
> | PCL Training Epoch | Prototype Matching Samples Accuracy (%) |
> | :--- | :--- |
> | 1 | 83.5% |
> | 20 | 85.0% |
> | 40 | 86.2% |
> | 60 | 86.5% |
> | 80 | 86.0% |
> | 100 | 86.8% |
>
> As training progresses, our Prototype Contrastive Learning (PCL) uses the initial matching results to guide the representation learning. This pulls semantically similar samples from different views closer in the latent space. The improved representations lead to better-defined prototypes, which in turn results in an increase in matching accuracy. In later stages, the model reaches a highly synergistic state where the accuracy stabilizes at a very high level.
>
> ## Reply to Questions (1):
>
> We sincerely thank the reviewer for this insightful and specific question.
>
> ### 1. Clarification on the MNN Construction Process:
>
> Our goal is to find high-quality, high-confidence positive pairs for contrastive learning. MNN is a powerful tool for this. In the mini-batch training, for a sample $x$ in View A, we first find its k-nearest neighbors in View B. Then, for each of these neighbors, we check if their k-nearest neighbors (back in View A) include $x$. A positive pair is formed only when this reciprocal relationship holds. This stringent strategy effectively filters out spurious connections caused by noise or distribution gaps, providing a reliable foundation for learning.
>
> ### 2. Handling the "Failure to Find MNN" Case:
> In our model, if a node fails to find any MNN pairs, our graph construction strategy automatically degrades to using its kNN pairs as the positive pairs. We prioritize the most reliable, reciprocal MNN connections. When these high-confidence links exist, we use them to provide the strongest supervisory signal. When such an ideal MNN link cannot be found, we do not want to abandon the data. Therefore, we use its kNN connections as a fallback. While kNN links have lower confidence than MNN, they still provide valuable neighborhood information that is far better than random, ensuring that every data point participates in the contrastive learning process and guaranteeing training stability.
>
> To visualize the number of samples where MNN is not found during the training process, we have investigated its convergence by reporting the loss value and the count of samples with undetected MNNs as training progresses in the supplementary material. Experiments are run on Caltech-101 with $k=5$ under both defined incompleteness scenarios. As depicted in the figures in the supplementary material, the loss value decreases sharply in the first 10 epochs before converging gradually. Regarding the number of samples with undetected MNNs, this count drops sharply within the first two epochs and subsequently remains at a low level throughout the training. Although samples with undetected MNNs persist, they constitute a very small proportion of the total dataset.
>
> ## Reply to Questions (2):
>
> We sincerely thank the reviewer for this insightful question, as it addresses a key design choice in our model. We employ these two different strategies because the two stages of our model have fundamentally different objectives.
>
> ### 1. Objective of Stage 1: To Obtain a Stable and Reliable Initial Prototype Estimation.
>
> In the first stage, our primary goal is to derive a stable and meaningful initial cluster structure from the instance embeddings, which have been pre-trained by GCL but are still relatively coarse.K-Means is a classic, robust, and effective algorithm for this task. It operates on a fixed feature space and provides us with a reliable, non-random initialization of prototypes. This high-quality initialization is the cornerstone for the success of the subsequent Prototype Graph Matching (PGM) stage. Using a learnable layer at this early stage would be problematic. The initial instance embeddings are not yet well-optimized, so a learnable layer would lack a stable and meaningful gradient signal, potentially leading to training instability.
>
> ### 2. Objective of Stage 2: To Perform End-to-End Joint Optimization.
>
> In the second stage, our goal shifts from simple initialization to jointly and synergistically optimizing both the instance representations and the cluster centers (prototypes). The learnable clustering layer is differentiable. As part of the network, it allows gradients from the clustering-related loss to be backpropagated to the feature encoders.

---

> > ### Comment · Reviewer_83hp · 2025-08-04
> >
> > Thank you for the authors’ detailed and well-organized rebuttal. I appreciate the thorough responses provided to each of the concerns.
> > I believe the key concerns have been sufficiently addressed. I will maintain my original score.

---

> > > ### Author Response · Authors · 2025-08-06
> > >
> > > Thank you again for your time and valuable feedback. We are pleased that our rebuttal and clarifications have addressed your concerns.

---

### Note · Authors · 2025-08-13

We are very grateful for the time and constructive feedback provided by the reviewers and the ACs. This process has been incredibly helpful.

Our responses have demonstrated that our core PGM component is theoretically grounded and that our use of contrastive learning is a unique paradigm for this correspondence-free problem. Furthermore, our new experiments confirm CPMN's robustness and its superior computational efficiency.

We hope that our detailed responses have successfully addressed the initial concerns. We are confident that the manuscript has been significantly strengthened by your feedback. Thank you once again for your time and careful consideration.

---

### Decision · Program_Chairs · 2025-09-17

**Decision:**

Accept (poster)

**Comment:**

After the rebuttal and discussion phases, the paper received scores of 5, 5, 4, and 4, exceeding the expected threshold for acceptance. The main concerns were the efficiency of CPMN, the rationale for using Mutual Nearest Neighbors, and the lack of theoretical explanation for key components like prototype matching. After the rebuttal, all of the reviewers' concerns were addressed.

Having reviewed the comments and the authors' responses, I believe the paper meets the acceptance criteria for NeurIPS.